# Effect of Slightly Acidic Electrolyzed Water Combined with Nano-Bubble Sterilization on Quality of *Larimichthys crocea* During Refrigerated Storage

**DOI:** 10.3390/foods14152754

**Published:** 2025-08-07

**Authors:** Jiehui Zhong, Hongjin Deng, Na Lin, Mengyao Zheng, Junjie Wu, Quanyou Guo, Saikun Pan

**Affiliations:** 1School of Ocean Food and Biological Engineering, Jiangsu Ocean University, Lianyungang 222005, China; jiehui010@163.com; 2East China Sea Fisheries Research Institute, Chinese Academy of Fisheries Sciences, Shanghai 200090, China; denghongjin1020@163.com (H.D.); lina903368043@163.com (N.L.); zhengmengyao_0210@163.com (M.Z.); wujj@ecsf.ac.cn (J.W.); 3College of Ocean Studies, Ningde Normal University, Ningde 352100, China

**Keywords:** large yellow croaker, microorganism, volatile flavor compound, storability, quality

## Abstract

The large yellow croaker (*Larimichthys crocea*) is susceptible to microbial contamination during storage due to its high protein and moisture contents. This study was designed to find a new way to reduce bacteria in large yellow croakers by combining slightly acidic electrolyzed water (SAEW) with nano-bubble (NB) technology. Exploring the effects of available chlorine concentration (ACC), processing time, and water temperature on the bacteria reduction effect of the SAEW-NB treatment for large yellow croakers. Also, the effects of the SAEW-NB combined treatment on sensory evaluation, total viable counts (TVCs), total volatile basic nitrogen (TVB-N), texture, taste profile, and volatile flavor compounds of large yellow croakers were analyzed during the storage period at 4 °C. The results show that the SAEW-NB treatment achieved significantly enhanced microbial reduction compared to individual treatments. Under the conditions of a 4 °C water temperature, 40 mg/L ACC, and 15 min treatment, the SAEW-NB treatment inhibited the increases in physical and chemical indexes such as TVC and TVB-N, maintained the fish texture, and delayed the production of off-flavor volatiles such as aldehydes, alcohols, esters, and ketones, compared with the control group (CG) during storage at 4 °C. In conclusion, the SAEW-NB treatment could better retard fish spoilage, extending the shelf life by approximately 2 days. It might be a promising new industrial approach for large yellow croakers’ storage.

## 1. Introduction

As one of the important aquaculture economic fish in China, large yellow croakers (*Larimichthys crocea*), also known as yellow croakers, cucumber croakers, etc., are mainly distributed in Fujian, Zhejiang, Shandong, and Guangdong [1]. The large yellow croaker has a reputation as a “National Fish” because of its delicious meat, rich in vitamins, trace elements such as selenium, phosphorus, potassium, and unsaturated fatty acids that are beneficial to the human body [2]. In recent years, the aquaculture industry of large yellow croakers in China had been growing rapidly, with aquaculture production reaching 281,000 tons in 2023 [3]. Nevertheless, large yellow croakers are susceptible to microorganisms, protein denaturation, and autolytic enzymes during storage, transportation, and marketing, which easily caused spoilage and deterioration and deteriorated their quality [4]. Traditional sales methods hardly meet the current market demands, making it crucial to explore novel preservation technologies.

Nano-bubbles (NB)s are spherical bubbles < 200 nm in diameter with properties such as strong oxidation and mechanical sterilization [5]. A highly reactive hydroxyl radical (–OH) is released from NBs upon rupture, which can disrupt microbial biofilms and other structures, thereby achieving a sterilizing effect [6,7]. Also, the pressure changes from NB rupture could have a direct mechanical damaging effect on microorganisms [8]. In *Vibrio parahaemolyticus* and *Listeria monocytogenes* biofilm removal studies, NBs could reduce microorganisms by one to three orders of magnitude [9]. When NBs were used to treat *Vibrio parahaemolyticus* in aquaculture water, the bacterial reduction rate was 31% [10]. Although NBs destroyed microbial biofilms and other structures, the bacterial reduction effect was not obvious when acting alone. It was shown that NB treatment alone was not effective at inhibiting *Vibrio parahaemolyticus* and *Aeromonas hydrophila* [11]. Combining the application of NBs with green bacterial reduction agents becomes an important direction to enhance the effectiveness of bacterial reduction.

Slightly acidic electrolyzed water (SAEW) is produced as a non-thermal sterilization technique by electrolysis of an aqueous solution of sodium chloride or hydrochloric acid [12]. Based on a low pondus hydrogenii, high oxidation reduction potential, and available chlorine, SAEW achieves sterilization by disrupting microbial cell membranes, inhibiting enzyme activities, and through oxidative reactions [13]. The efficient and broad-spectrum, non-residue characteristics of SAEW have been used widely in the field of aquatic product preservation. In *Thunnus obesus* [14], *Teuthida* [15] and *Pampus* [16] studies, SAEW treatment reduced the number of microorganisms on the surface of aquatic products by 2~3 logarithmic levels, significantly inhibited the growth of total viable counts, TVB-N, and TBA values during the storage period, and prolonged the shelf-life by 2~3 days, while maintaining a better organoleptic quality. Meanwhile, the effect of SAEW, combined with NB treatment, on the sensory quality, microbial changes, taste, and volatile flavor compounds of vacuum-packed large yellow croaker during refrigerated storage has rarely been reported on.

In this experiment, large yellow croakers were treated with SAEW combined with NBs and then vacuum-packed at 4 °C for refrigeration. By optimizing the parameters of the SAEW-NB treatment, this study systematically evaluated its effects on the physicochemical properties (TVB-N, TBA, and pH), microbial communities (H_2_S-producing bacteria, *Pseudomonas*, and *Enterobacteriaceae*), taste, and volatile flavor compounds of large yellow croakers. And combined with correlation analysis to reveal the trend of quality change in large yellow croaker. This study provided critical theoretical support and practical guidance for the research and development of preservation technologies for aquatic products.

## 2. Materials and Methods

### 2.1. Large Yellow Croaker

At night, farming large yellow croakers were caught in the sea of Sanduwan, Ningde, China. Then, large yellow croakers were placed on ice and immediately transported by refrigerated truck within 2 h to Cai Shi Aquatic Co. (Ningde, China). A total of 136 large yellow croakers were selected and transported to the laboratory within 1 h in stratified ice–fish layers. The scales, gills, and viscus of the large yellow croakers were removed and cleaned with fresh water within 2 h [17]. The total elapsed time between capture and treatment was 5 h. Of the selected fish, 16 samples were for comparative experiments, 72 samples were for single-factor experiments, and 48 samples were randomly selected to be divided into the control group (CG) and the slightly acidic electrolyzed water combined with nano-bubbles treatment group (SAEW-NB). After vacuum packing, the large yellow croakers were stored in a low-temperature incubator (4 ± 1 °C) (HWS-70B, Taisite Instrument Co., Tianjin, China) and sampled at 0, 2, 4, 6, 8, and 10 days, respectively. The basic information and composition of the large yellow croaker (Table 1) were determined with reference to the methods described by Deng et al. [18].

### 2.2. Preparation of SAEW and NB

The SAEW-NB treatment system was designed with reference to the experimental approaches of Lan et al. [19] and Jhunkeaw et al. [20].

SAEW: The SAEW of different available chlorine concentrations (ACCs) was generated by adjusting the electric current magnitude of the electrolyzed water generator (Harmong-1000, Rui Andre Environmental Equipment Co., Ltd., Beijing, China) to electrolyze a solution containing 5% sodium chloride (NaCl). This was pre-cooled for storage away from the light. Before the experiment, the pondus hydrogenii (pH), oxidation reduction potential (ORP), and ACC were measured by using a pH meter (PHS-3E, Inase Scientific Instrument Co., Ltd., Shanghai, China), ORP meter (ORP-BL, Qiwei Instrument Co., Ltd., Hangzhou, China), and ACC meter (GDYS-104SM, Jida-Little Swan Instrument Co., Ltd., Changchun, China), respectively.

NB: The NBs were generated by using a nano-bubble generator (HM-KS-3G, Huankong Research Institute, Kaifeng, China). The average diameter of the NBs was <200 nm, and the density was 1 × 10^8^ particles/mL. The pump flow rate of the nano-bubble generator was 5 L/min. The pump was connected to an acidic electrolyzed water tank with a diameter of 50 cm and a capacity of 30 L.

SAEW-NB operation is illustrated in Figure 1. Firstly, SAEW with the target available chlorine concentration was generated using an electrolyzed water generator, transferred to a holding tank, and pre-cooled to the required water temperature. Secondly, the nano-bubble generator was connected. Its pump drew water from the electrolyzed water tank, then mixed purified air and SAEW in the nano-bubble generator to produce SAEW-NB water. The SAEW-NB water flowed back into the water tank. Finally, rinse and vacuum pack.

### 2.3. Comparative Experiments

Sixteen large yellow croakers were randomly divided into a CG, SAEW, NB, and SAEW-NB group, with four samples in each group. Referring to the results of Jin et al. and Liu et al. [21,22], the ACC was taken as 40 mg/L, solid–liquid ratio of 1:6 and the processing time was 15 min for the pre-experiment (Table 2).

**Table 2 foods-14-02754-t002:** Conditions of different treatment methods.

Conditions	Groups			
	CG	SAEW	NB	SAEW-NB
ACC (mg/L)	-	40	0	40
Solid—liquid (g: mL)	-	1:6	1:6	1:6
Processing time (min)	-	15	15	15
Water temperature (°C)	-	RT	RT	RT

Note: “-“ represents the group without any treatment. Available chlorine concentration (ACC). Control group (CG). Slightly acidic electrolyzed water (SAEW). Nano-bubbles (NB). Slightly acidic electrolyzed water combined with nano-bubbles (SAEW-NB). Room temperature (RT).

### 2.4. Single-Factor Experimental Design for SAEW-NB

The condition of a single factor is shown in Table 3. The sterilization effect of single factor treatment was analyzed by setting ACC, processing time, and water temperature as single factor according to the solid–liquid ratio of 1:6. When a single factor was analyzed, the remaining factors were taken as fixed values, and total viable counts (TVCs) were determined for the SAEW-NB and CG.

In the present experiment, the actual ACC, pH, and ORP results of SAEW-NB are shown in Table 4.

### 2.5. Microbiological Enumeration

Referring to GB 4789.2-2022 [23] for microbiological analysis, total viable counts, *Pseudomonas*, H_2_S-producing bacteria, and *Enterobacteriaceae* were counted (colony counter, SN-JLQ, Shangpu Instrument Equipment Co., Ltd., Shanghai, China) using plate count agar, *Pseudomonas* CFC selective agar, iron agar, and violet red bile glucose agar, respectively. Microorganisms of *Enterobacteriaceae* were placed in a constant-temperature incubator at 37 °C for 48 h, and the rest of the microorganisms were placed in a constant-temperature incubator at 30 °C for 72 h for colony counting. The results were expressed as log_10_ CFU/g.

### 2.6. Measurement of pH

After mashing skinless back muscle (5.0 g), distilled water was added and filtered after standing for 30 min. The supernatant was taken and measured three times in parallel using a pH meter (PHS-3E, Inase Scientific Instrument Co., Ltd., Shanghai, China).

### 2.7. Measurement of Total Volatile Base Nitrogen (TVB-N)

The content of TVB-N was determined by the semi-micro method with reference to the Determination of Total Volatile Basic Nitrogen in Foods (GB 5009.228-2016 [24]).

### 2.8. Measurement of 2-Thiobarbituric Acid (TBA)

The skinless back muscle (5.0 g) was weighed and placed in a 150 mL conical flask. Then, the 25 mL trichloroacetic acid mixed solution (75 g/L, 0.1% EDTA) was added to the conical flask and shaken for 30 min. The sample mixture was centrifuged at 4 °C and 4500 r/min for 15 min. The supernatant (5 mL) was mixed with 2-thiobarbituric acid solution (5 mL) and heated in a boiling water bath for 30 min. After temperature reduction, the absorbance values were determined by spectrophotometer at 532 nm and 600 nm, respectively, three times in parallel for each sample. Based on the formula
$TBA\;(mg/kg)=\frac{\left(A_{532}-A_{600}\right)}{155\times 5}\times 72.06\times 0.05\times 100$, the TBA value was calculated. Here, A_532_ and A_600_ were the absorbance values at wavelength 532 nm and 600 nm, respectively.

### 2.9. Measurement of Texture

The skinless back muscles (length × width × height = 2 cm × 2 cm × 1 cm) were measured by using texture profile analysis (TPA mode, TMS-PRO, Food Technology Corporation Co., Sterling, TX, USA). It was equipped with a P/5 probe with a test speed of 50 mm/min, a deformation of 50%, and a return distance of 30 mm. The muscles in each group were measured in parallel four times. The indexes included hardness, cohesiveness, springiness, gumminess, and chewiness.

### 2.10. Measurement of Taste Profile

After shaking and mixing 50 g of stirred back muscle with 250 mL of ultrapure water, the mixture was centrifuged at 4 °C and 4500 r/min for 15 min. The supernatant was filtered and then transferred to a special container for the electronic tongue (SA-402B, INSENT Co., Tokyo, Japan). The electronic tongue was configured with 5 lipid membrane sensors (AAE, CT0, C00, AE1, and GL1, INSENT Co., Tokyo, Japan) and 2 Ag/AgCl reference electrodes (INSENT Co., Tokyo, Japan). The sensors are designed to detect 5 taste signal values of sample solutions, including umami, saltiness, bitterness, astringency, and sweetness, and 3 aftertaste signal values, namely umami aftertaste, bitter aftertaste, and astringent aftertaste. Each sample was set up for 4 cycles and the last 3 were selected for analysis. For the determination of sweetness, each sample was set to measure 5 cycles, and the last 3 were taken for analysis.

### 2.11. Measurement of Volatile Flavor Compounds

Sample processing: After the back muscle was stirred, 3 g of back muscle was weighed into a 20 mL headspace bottle.

Automatic headspace injection: The injection volume was 200 μL, the injection needle temperature was 85 °C, the incubation temperature of the headspace bottle was 40 °C, the incubation time was 20 min, and the incubation speed was set at 500 r/min.

Gas chromatography (GC) conditions: A MXT-WAX column (30 m × 0.53 mm, 1 μm) was used, the column temperature was 60 °C, high-purity nitrogen (N_2_, purity ≥ 99.999%) was used as the carrier gas, and the column flow rate was as follows: the initial column flow rate was 2 mL/min, held for 2 min, increased linearly to 100 mL/min from 2 min to 10 min, and to 150 mL from 10 min to 20 min.

Ion mobility spectrometry (IMS) conditions: The length of the drift tube was 98 mm, the temperature of the drift tube was 45 °C, the linear voltage inside the drift tube was 500 V/cm, the flow rate of the drift gas was 150 mL/min, and the drift gas was N_2_.

### 2.12. Sensory Evaluation

The sensory evaluation table is formulated according to GB/T 18108-2024, SC/T 3101-2010 and SC/T 3216-2016 (Table 5) [25,26,27]. Large yellow croakers were removed from a 4 °C low-temperature incubator, rinsed with fresh water, and their mid-sections were steamed in boiling water for 8 min. A group of eight professionally trained evaluators (20–30 years old, 4 males and 4 females) were selected to assess the appearance, flavor, morphology, and texture of large yellow croakers based on the sensory evaluation table criteria (water as a neutralizer). For these, the score of 8 to 10 was high quality, the score of 4 to 8 was medium quality, and the score of 2 to 4 was low quality.

### 2.13. Statistical Analysis

The results are expressed as mean ± standard deviation (SD). Microsoft Excel 2021 and IBM SPSS Statistics 21.0 were used to analyze the data. *p* ≤ 0.05 indicates significant differences. GraphPad Prism 10.0 was used to perform drawing editing. The GC-IMS was used to characterize the characteristic flavor compounds using the built-in NIST database and IMS database, and the fingerprints of volatile flavor compounds were constructed using the Gallery plug-in. Principal component analysis (PCA) was performed using MetaboAnalyst 6.0 (https://www.metaboanalyst.ca/) (accessed on 17 May 2025).

## 3. Results and Discussion

### 3.1. Sterilization Effects of CG, NB, SAEW, and SAEW-NB

To a certain extent, TVC could predict the shelf life of food products. In Figure 2, compared with the CG, the TVC was reduced by 0.19, 0.56, and 1.03 log_10_ CFU/g in the NB, SAEW, and SAEW-NB groups, respectively, with reduction rates of 35.20%, 72.22%, and 91.11%. And the SAEW and SAEW-NB groups had a significant bacterial reduction effect (*p* < 0.05). In addition, the bacterial reduction effect was significantly better in the SAEW-NB group than in the other groups (*p* < 0.05). On the one hand, the water flow was increased by the pressure difference generated by the rupture of air bubbles, which promoted the uniform distribution of available chlorine on the body surface of the fish and in the interstitial space of the tissues [11]. On the other hand, the nanoscale mass transfer pathway could break through the biofilm barrier and enhance the killing efficiency of microorganisms in deep tissues [7]. The results indicated that the SAEW-NB co-treatment significantly enhanced the bacterial reduction efficacy through the dual mechanism of physical perturbation (nano-bubble implosion) and chemical sterilization (available chlorine oxidation). It was compatible with the combination effect of composite bacterial reduction technologies in food processing.

### 3.2. The Results of Single-Factor Experiments for SAEW-NB

The experiments were performed at a water temperature of 4 °C, a processing time of 10 min, and a solid–liquid ratio of 1:6 according to the ACC gradient designed in Table 2 As shown in Figure 3A, the TVC in the SAEW-NB group showed a decreasing trend with the increase in ACC. When the ACC was greater than 40 mg/L, the TVC in the SAEW-NB group tended to level off. Studies in Oysters (*Ostreidae*) [28] and Chinese shrimp (*Fenneropenaeus chinensis*) [29] found a decreasing trend in TVC after SAEW treatment with increasing of ACC. When ACC was greater than 60 mg/L or 80 mg/L, the TVC tended to level off after SAEW treatment. This was probably due to the differences in the dominant flora in different aquatic products. Nevertheless, the effective bacterial reduction components of SAEW were susceptible to factors such as light and time. Thus, the ACC of 40 mg/L was selected as the subsequent experimental parameter by combining the effect of bacterial reduction and the feasibility of actual production.

The experiments were conducted at a water temperature of 4 °C, an ACC of 40 mg/L, and a solid–liquid ratio of 1:6 according to the processing time gradient designed in Table 2 As shown in Figure 3B, the TVC in the SAEW-NB group showed a decreasing trend with increasing treatment time. When the processing time was greater than 15 min, the TVC tended to level off. The results were similar to those of Chinese shrimp treated with SAEW [29]. In the open treatment environment, factors such as light and gas–liquid exchange accelerated the depletion of available chlorine components in SAEW, which affected the effect of treatment for a long time [30]. Consequently, the treatment time of 15 min was selected by combining the bacterial reduction effect with the operational costs (equipment turnover rate and energy consumption) in actual production.

The experiments were carried out at a treatment time of 15 min, an ACC of 40 mg/L, and a solid–liquid ratio of 1:6 according to the water temperature gradient designed in Table 2, and Figure 3C shows that the TVC of the SAEW-NB group was significantly lower than that of the CG at 2–10 °C, which indicates that the low temperature had an effective bacterial reduction effect. Research has shown that ozone ice and other bacteriostatic agents in the use of the process also reduce the temperature to slow down the rate of decay of the ozone activity, to achieve better bacteriostatic effects [31]. Although 2 °C demonstrated superior sterilizing efficacy compared to other temperatures, studies indicate that this condition promotes the formation of large ice crystals in muscle cells, which can break the structure of the cell membrane, which leads to the loss of juice [32]. At the same time, this excessively inhibits the activity of ATPase, which slows down the process of autolysis of the muscle and makes the meat of the finished product hard [33]. While there was no significant difference in the bacterial reduction effect of water temperatures of 4–10 °C, studies have revealed that oxidase activity at 6–10 °C was higher than at 4 °C, resulting in accelerated oxidation rates under elevated temperatures [34]. Importantly, 4 °C is a common food cold chain base temperature [35]; processing at a water temperature of 4 °C can balance the sterilization effect of the SAEW-NB system and fish quality, and can seamlessly connect to the cold chain system to avoid temperature fluctuations and quality loss.

In conclusion, the SAEW-NB bacterial reduction process is obtained for large yellow croakers at a solid–liquid ratio of 1:6, an active chlorine concentration of 40 mg/L, treatment for 15 min, and a water temperature of 4 °C.

### 3.3. Sensory Evaluation Analysis

As one of the core dimensions of food quality assessment, the results of sensory evaluation are directly related to consumer acceptance and market feedback [36]. For this experiment, the sensory score of the SAEW-NB group (9.69 points) was lower than that of the CG (9.81 points) on day 0 of storage (Table 6). This was probably related to the residual odor of the SAEW-NB treatment. One study demonstrated a decreasing trend in initial sensory score (day 0) due to the residual chlorine odor when raw Atlantic salmon (*Salmo salar*) filets were treated with the SAEW combined ultraviolet and ultrasonic technologies [37]. In the present study, the sensory scores of the SAEW-NB group and the CG were above 8 on days 0 and 2 of storage (Figure 4), indicating the freshness of the large yellow croakers. During storage up to the 8th day, the sensory scores of the SAEW-NB group and CG showed a decreasing trend with the phenomenon of fishy and amine flavors, which could be related to the metabolites produced by microbial activities. But the sensory score of the SAEW-NB group was higher than that of the CG, indicating that the SAEW-NB treatment could inhibit the growth of odor-producing microorganisms. During the whole storage period, the sensory score of the SAEW-NB group was higher than that of the CG, which was consistent with the trends of TVB-N and TVC, indicating that the SAEW-NB treatment was effective in terms of improving the odor, maintaining quality, and reducing the sensory loss during the storage period.

### 3.4. Microbiological Analysis

Microorganism growth and reproduction is one of the main causes of food spoilage [38]. During the present study, the large yellow croaker showed a logarithmic increase in TVC in the CG and the SAEW-NB group during storage at 4 °C (days 0–10) (Table 7). On day 10, TVC reached 7.02 ± 0.06 log_10_ CFU/g in the CG, while TVC did not reach 7 log_10_ CFU/g in the SAEW-NB group. Research has indicated that a fish TVC less than 7 log_10_ CFU/g is the threshold of microbial safety [39]. This indicates that the SAEW-NB treatment could significantly delay the microbial growth process of large yellow croakers and provide a more powerful guarantee of fish freshness. In addition, *Pseudomonas*, *Shewanella*, and *Enterobacteriaceae* were the dominant spoilage bacteria responsible for the deterioration of fish meat, the development of fishy odors, and the destruction of texture during the low-temperature storage period (4 °C) [40]. Within this experiment, the number of *Pseudomonas*, H_2_S-producing bacteria, and *Enterobacteriaceae* increased in the CG and the SAEW-NB group as the duration of the storage period increased. On days 0–10, the numbers of *Pseudomonas*, H_2_S-producing bacteria, and *Enterobacteriaceae* were lower in the SAEW-NB group than in the CG. This indicated that SAEW-NB treatment had an inhibitory effect on *Pseudomona*, H_2_S-producing bacteria, and *Enterobacteriaceae*, and delayed the spoilage of fish muscle and contributed to maintaining the quality and safety of the product.

### 3.5. The Analysis of pH, TVB-N, and TBA

The pH, TVB-N, and TBA values were important indicators of food quality assessment, which together could reflect the degree of spoilage and quality change in the food products [41]. Firstly, in the course of this study, the pH values of the CG and the SAEW-NB group showed a decreasing and then increasing trend with the prolongation of storage time (Figure 5A). On day 0, the pH of the SAEW-NB group was lower than that of the CG, which could be attributed to the fact that the SAEW-NB treatment adsorbed H^+^ from the surface of the fish and the superficial tissues, creating a transient slightly acidic environment [42]. On days 0–2, the reason for the pH decreases in the CG and the SAEW-NB group might be related to the production of lactic acid from glycogenolysis and phosphocreatine catabolism to produce phosphoric acid in the fish after death [43]. On days 2–10, the reason for the consistently elevated pH in the CG and the SAEW-NB group could be the depletion of glycogen in the fish and the microorganisms that have started to degrade proteins and produce alkaline substances. Interestingly, on days 2–10, the rate of pH rise was higher in the CG than in the SAEW-NB group. This may originate from the inhibitory effect of SAEW-NB treatment for dominant spoilage bacteria such as *Pseudomonas* and *Shewanella*, inhibiting their metabolic process of degrading proteins to produce alkaline products, thus slowing down the rate of pH increase.

Secondly, on day 0, the TVB-N values of both the CG and the SAEW-NB group were below 10 mg/100 g, indicating that the large yellow croakers maintained good quality. (Figure 5B). With the extension of storage time, the TVB-N values of both the CG and the SAEW-NB group showed an increasing trend, and the rate of increase in TVB-N values of the SAEW-NB group was lower than that of the CG. On day 4, the TVB-N value of the CG was 15.89 mg/100 g, which exceeded the limit value for superior grade products specified in GB/T 18108-2024 [25], while that of the SAEW-NB group was 9.39 mg/100 g. On day 10, the TVB-N value of the CG was 41.30 mg/100 g, which exceeded the conformity limit, while the SAEW-NB group was 22.37 mg/100 g. This demonstrated that SAEW-NB treatment could effectively reduce the production of nitrogenous substances such as ammonia and amines in fish. This result was consistent with the trend of total viable counts and sensory scores, which confirmed that SAEW-NB treatment could effectively slow down the rate of increase in TVB-N.

Finally, in our experiment, the TBA values of both the CG and the SAEW-NB group tended to increase with longer storage time (Figure 5C). On day 0, TBA values were higher in the SAEW-NB group than in the CG. This was probably triggered by the mild oxidation of fat within the fish triggered by the –OH produced by the rupture of the strongly oxidizing components of SAEW with NBs, resulting in a transient increase in TBA values. Consistent results were found for SAEW combined with ultrasound treatment of tuna [14]. The rate of increase in TBA values was consistently higher in the CG than in the SAEW-NB group from days 2–10. This demonstrated that the SAEW-NB treatment was effective at slowing down the accumulation of fat oxidation products (aldehydes and ketones) in fish, thus slowing down the spoilage process.

### 3.6. Textural Analysis

As core food quality indicators, textural parameters (hardness, springiness, chewiness, gumminess, and cohesiveness) directly reflect aquatic product freshness [44]. During the present study, the hardness and springiness of the SAEW-NB were marginally lower than those of the CG (3.89% and 2.55%) on day 0 (Figure 6). It is possible that the low-pH environment and available chlorine concentration of SAEW-NB, as well as the oxidation of –OH led to the breakage of intermolecular disulfide bonds of myofibrillar proteins, triggering mild damage to the protein structure [45]. This is consistent with research findings that show that the hardness and springiness of large yellow croakers treated with acidic electrolyzed water are slightly lower than those of the control group (5.31% and 10.37%) [46].

During storage, all parameters except cohesiveness declined progressively. On days 0–10, the decrease rates of hardness, springiness, gumminess, and chewiness were higher in the CG (46.23%, 26.10%, 48.73%, and 33.26%) than in the SAEW-NB group (24.27%, 19.94%, 33.93%, and 20.29%), suggesting that the SAEW-NB treatment maintained the structural integrity of myofibrils in the muscle [47]. Long-term oxidative damage during refrigerated storage impacted texture deterioration more substantially than the initial minor changes from SAEW-NB treatment. In addition, the cohesiveness of the SAEW-NB group and CGs showed small changes in the range of 0.30 to 0.50 with no trend. This indicates that SAEW-NB treatment had less of an effect on muscle intercellular adhesion. These results are consistent with findings on cohesiveness stabilization in golden pomfret filets [48].

### 3.7. Taste Profile Analysis

Electronic tongues mimic human taste through biomimetic sensor arrays that detect key taste (sourness, sweetness, bitterness, saltiness, umami, and astringency) and convert responses into quantitative metrics. This provides unbiased taste assessment critical for food quality control [49]. Within this experiment, the e-tongue data of the fish from the CG and SAEW-NB group on days 0–10 were subjected to principal component analysis. The results are shown in Figure 7; the cumulative contribution rate of the first two principal components (PC1 and PC2) exceeded 85% throughout days 0–10. This demonstrates their efficacy in characterizing the overall flavor profile of the samples while revealing persistent intergroup taste differences at each time point. Additionally, parsing the e-tongue data by radargram revealed that the main differences between the CG and the SAEW-NB group were focused on sweetness and freshness signals (Figure 8). On day 0, the sweet and umami flavors’ signal intensities were 6.67% and 10.86% lower in the SAEW-NB group compared to the CG, respectively. This was probably related to the oxidation of free amino acids (for example, glutamic acid and aspartic acid) in the samples and concomitant degradation of fresh-taste compounds, such as nucleotides [50]. This result was similar to the slight loss of fresh taste compounds in the SAEW treatment of tilapia filets [51]. The magnitude of the sweetness change in the CG was greater than that in the SAEW-NB group throughout the storage process, indicating that the SAEW-NB treatment had the effect of maintaining the sweetness of the fish.

### 3.8. Analysis of Volatile Flavor Compounds

The large yellow croaker is rich in highly unsaturated fatty acids, which are easily oxidized and decomposed to become aldehydes, alcohols, ketones, and esters during storage. This oxidative degradation is a primary factor contributing to the deterioration of fish flavor and the loss of freshness [52]. In this experiment, to analyze the trend of volatile flavor compounds after CG and SAEW-NB treatment of the large yellow croaker, two groups of fish with different refrigeration times were examined. The results are shown in Table 8 and Figure 9, and a total of 47 compounds were detected, including 10 esters, 9 alcohols, 7 ketones, 6 aldehydes, 3 acids, and 12 alkanes and heterocyclic compounds.

The aldehydes, primary products of the oxidation of unsaturated fatty acids, serve as established early markers of freshness loss [53]. On days 0–2, key aldehydes—nonanal, butanal, hexanal, and 2,4-heptadienal—showed stronger heatmap signals in the CG versus the SAEW-NB treatment (Figure 9D, red box). This pattern points to SAEW-NB effectively slowing unsaturated fatty acid oxidation in fish tissue. As storage progressed, aldehyde signals weakened in both groups—driven by the depletion of fatty acid precursors and aldehydes converting to carboxylic acids or condensing into ketones [54].

In terms of alcohols, during the refrigerated storage, the CG and the SAEW-NB group exhibited time-dependent increases in concentrations of 1-butanol, 2-butanol, and 1-propanol. Heatmap signals in Figure 9B reveal significantly lower accumulation rates of these compounds in the SAEW-NB group compared to the CG. Research has demonstrated that 1-propanol and 2-methyl-1-butanol are metabolic products of yeast [55,56]. These findings suggest the enhanced metabolic function of yeast communities in the CG samples, whereas SAEW-NB treatment suppressed such microbial metabolism through its antimicrobial efficacy. Critically, 2-butanol—a diagnostic marker of mid- to late-stage spoilage, which is primarily generated by *Pseudomonas* and *Enterobacteriaceae* via amino acid degradation or fatty acid β-oxidation pathways [57]. The significantly reduced 2-butanol levels in the SAEW-NB groups further validate the capacity of SAEW-NB to inhibit metabolic activity of core spoilage microorganisms.

Among esters, Figure 9A reveals negligible heatmap signals for most detected esters in both the CG and the SAEW-NB group on day 0. Strikingly, intense heatmap signatures for pivotal esters—including ethyl acetate, ethyl 2-methylbutyrate, methyl-2-methylbutyrate, ethyl-butyrate, ethyl acetate, propyl propanoate, and methyl-2-methyl butyrate—appeared substantially earlier in the CG than in its SAEW-NB counterparts. This accelerated signal emergence reflects intensified lipid oxidation and microbial metabolism in the CG samples. Critically, studies have established that specific *Pseudomonas* species aerobically generate esters such as ethyl acetate and ethyl propanoate, thereby demonstrating SAEW-NB’s inhibitory effect on this genus [58].

Among ketones, days 8–10, the heatmap signals intensified for 2-butanone, 2-heptanone, and 3-hydroxy-2-butanone with significantly higher intensity in the CG compared with the SAEW-NB group (Figure 9C, yellow box). This observation aligns with established correlations between ketone accumulation and *Shewanella*/*Pseudomonas* proliferation [59,60]. Critically, elevated 2-heptanone concentrations induce rancid/musty off odors [61], while 3-hydroxy-2-butanone—a signature metabolite from leucine/isoleucine degradation that peaks prior to TVB-N exceeding thresholds—serves as a shelf life predictor [62]. The delayed 3-hydroxy-2-butanone accumulation in the SAEW-NB groups confirms its preservation efficacy. Furthermore, elevated 4-methyl-3-penten-2-one (a product of deep lipid oxidation [63]) signals in the CG at day 8 evidenced SAEW-NB’s lipid oxidation suppression.

Regarding alkanes and heterocyclic compounds, throughout storage, the CG samples exhibited stronger heatmap signals than their SAEW-NB counterparts (Figure 9E). This pattern stems from SAEW-NB’s dual inhibition of lipid peroxidation and microbial activity, effectively constraining the formation of hydrocarbon, furan, and pyrrole derivatives.

### 3.9. Correlation Analysis

The correlation analysis could provide a systematic perspective on the quality of aquatic products by revealing the mechanism of multi-factor synergy through the linear association between quantitative variables. Figure 10 shows that all indicators were correlated, and that sensory indicators were strongly positively correlated with microbial indicators such as TVC and *Enterobacteriaceae*, indicating that the proliferation of microbial populations directly affected the deterioration of sensory quality. TVB-N, TBA, and pH were positively correlated with microbial indicators, confirming that microbial activity was the main cause of proteolysis, lipid oxidation, and alkaline compound accumulation. In contrast, textural indicators such as hardness and elasticity were negatively correlated with microbial and other physicochemical indicators, revealing the destructive effect of spoilage bacteria proliferation and their metabolites on the muscle fiber structure and extracellular matrix [47].

## 4. Conclusions

This research optimized the parameters for combining SAEW with NB, and their effect on the preservation of large yellow croaker during storage. The results demonstrate that this novel treatment slowed down the pH value increase; reduced the generation of TVB-N and TBA; maintained better texture, taste, and sensory qualities; and effectively mitigated spoilage compared with the CG. In addition, the combined treatment demonstrated potent bacterial inhibition, keeping the total viable count within acceptable limits by day 10 of storage and effectively suppressing spoilage-related bacterial counts, such as *Pseudomonas* and H_2_S-producing bacteria. These findings confirm that SAEW-NB’s physico-chemical sterilization mechanism enhances product quality and extends shelf life by approximately 2 days under refrigerated conditions. The optimized parameters provide critical, theoretical, and operational foundations for developing efficient green preservation technologies in aquatic products, with demonstrated potential to drive technological innovation in industrial seafood processing.

## Figures and Tables

**Figure 1 foods-14-02754-f001:**
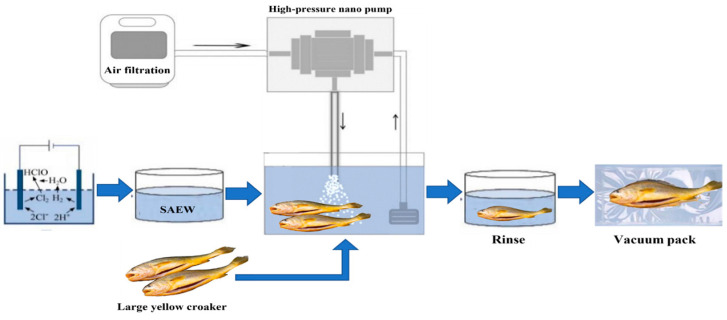
The sample production process.

**Figure 2 foods-14-02754-f002:**
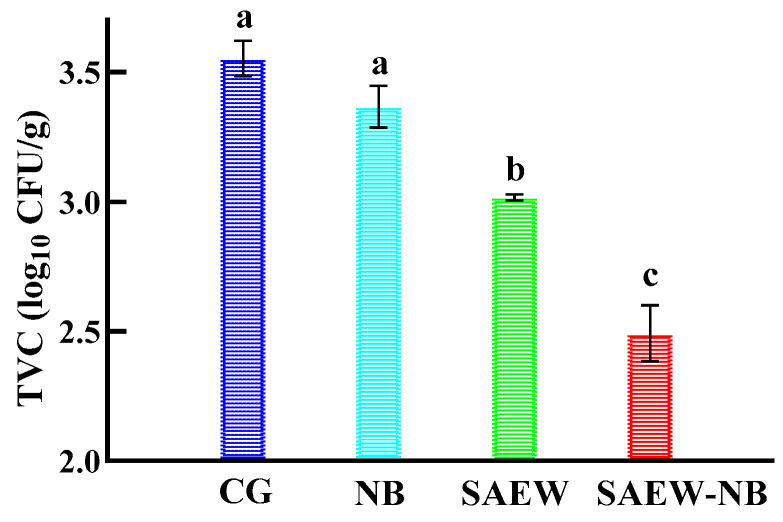
The TVC under different treatment methods. Note: (a–c) Values are shown from largest to smallest. a–c On the same row, different targets have significant values (*p* < 0.05). Total viable counts (TVC). Control group (CG). Nano-bubbles (NB). Slightly acidic electrolyzed water (SAEW). Slightly acidic electrolyzed water combined with nano-bubbles (SAEW-NB).

**Figure 3 foods-14-02754-f003:**
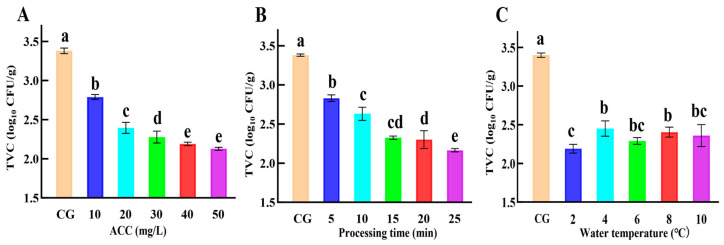
The TVC under different factors. Note: (a–e) Values are shown from the maximum to the minimum. a–e In the same row different targets had significant values (*p* < 0.05). Total viable counts (TVC). Control group (CG). Available chlorine concentration (ACC). (**A**–**C**) were the TVC of ACC, processing time and water temperature, respectively.

**Figure 4 foods-14-02754-f004:**
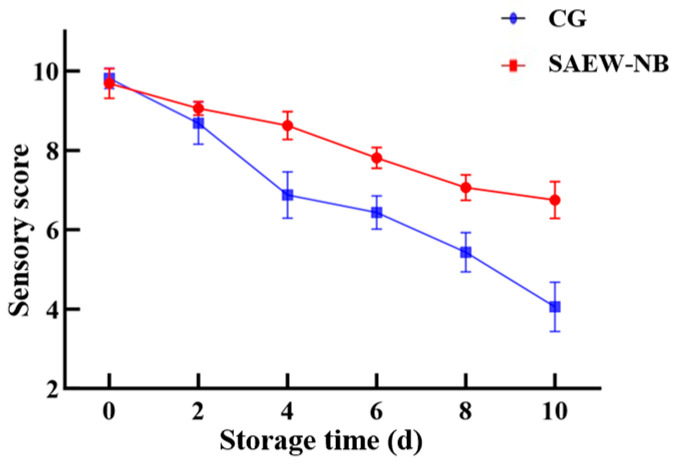
Sensory scores at different times. Note: Control group (CG). Slightly acidic electrolyzed water combined with nano-bubbles (SAEW-NB). Days (d).

**Figure 5 foods-14-02754-f005:**
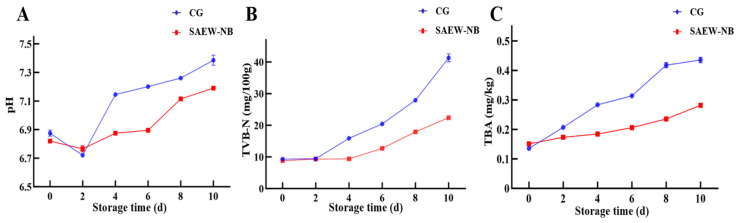
The pH, TVB-N, and TBA values at different periods. Note: (**A**)—Pondus hydrogenii (pH). (**B**)—Total volatile basic nitrogen (TVB-N). (**C**)—2-thiobarbituric acid (TBA). Control group (CG). Slightly acidic electrolyzed water combined with nano-bubbles (SAEW-NB). Days (d).

**Figure 6 foods-14-02754-f006:**
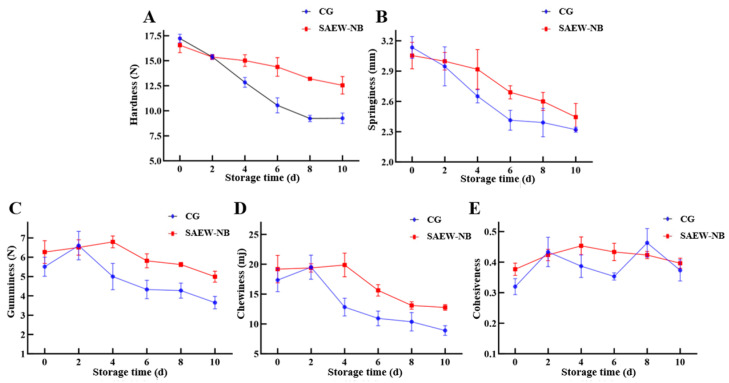
Effects of different treatments on textural analysis of large yellow croakers. Note: (**A**–**E**) are hardness, springiness, gumminess, chewiness, and cohesiveness, respectively. Control group (CG). Slightly acidic electrolyzed water combined with nano-bubbles (SAEW-NB). Days (d).

**Figure 7 foods-14-02754-f007:**
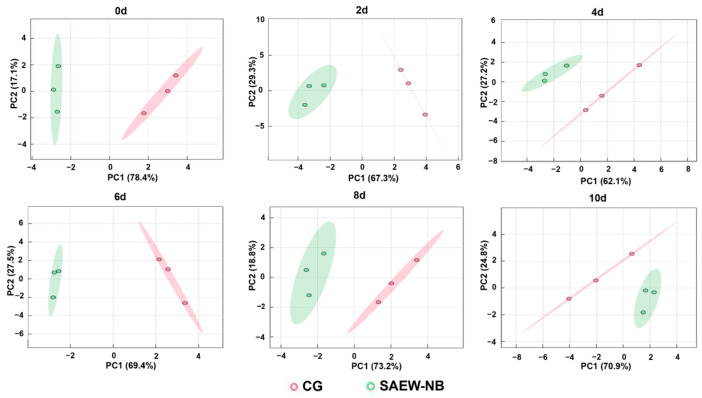
Principal component analysis (PCA) at different periods. Note: Control group (CG). Slightly acidic electrolyzed water combined with nano-bubbles (SAEW-NB). Days (d).

**Figure 8 foods-14-02754-f008:**
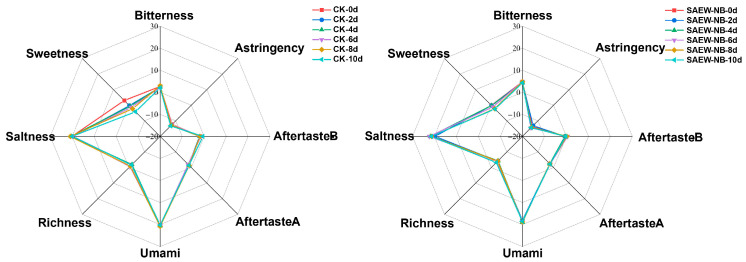
Radar charts in different periods. Note: Control group (CG). Slightly acidic electrolyzed water combined with nano-bubbles (SAEW-NB). Days (d).

**Figure 9 foods-14-02754-f009:**
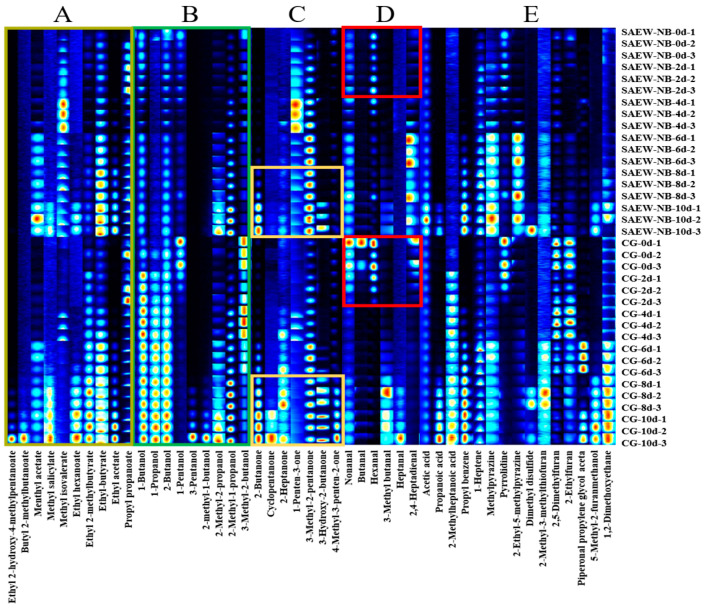
Fingerprint of volatile flavor compounds. Note: Region (**A**) represents ester compounds. Region (**B**) represents alcohol compounds. Region (**C**) represents ketone compounds. Region (**D**) represents aldehyde compounds. Region (**E**) represents alkane, furan, pyrrole, and other compounds. Control group (CG). Slightly acidic electrolyzed water combined with nano-bubbles (SAEW-NB). Three parallel samples per period.

**Figure 10 foods-14-02754-f010:**
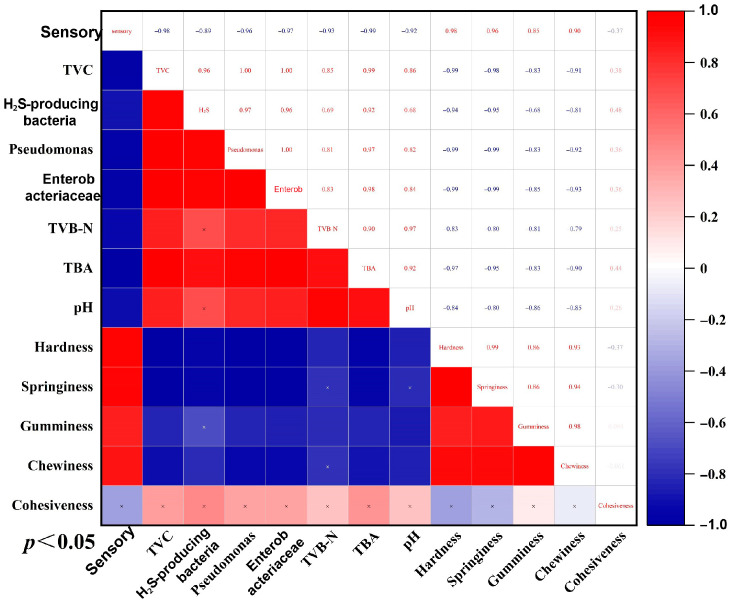
Correlation analysis among various indicators. Note: Total viable counts (TVC). Total volatile basic nitrogen (TVB-N). 2-thiobarbituric acid (TBA). Pondus hydrogenii (pH). *p* < 0.05 indicated significant difference. Red indicated a positive correlation. Blue indicated a negative correlation.

**Table 1 foods-14-02754-t001:** Basic information and composition of large yellow croaker.

Item	Moisture (%)	Crude Protein (%)	Crude Lipid (%)	Crude Ash (%)	Body Weight (g)	Body Length (cm)
Large yellow croaker	72.23 ± 2.67	17.83 ± 2.63	6.85 ± 1.23	1.61 ± 0.46	350 ± 50	30 ± 2

**Table 3 foods-14-02754-t003:** Condition of single factor.

Levels	Factors		
	ACC (mg/L)	Processing Time (min)	Water Temperature (°C)
1	10	5	2
2	20	10	4
3	30	15	6
4	40	20	8
5	50	25	10

Note: Available chlorine concentration (ACC). The ORP ranges corresponding to ACC are 600–700, 650–750, 700–800, 750–850, and 800–900 mv, respectively.

**Table 4 foods-14-02754-t004:** Parameters of SAEW-NB.

Levels	Factors		
	ACC (mg/L)	pH	ORP (mv)
1	10.9 ± 0.70	6.07 ± 0.05	881.7 ± 5.00
2	23.0 ± 0.06	5.93 ± 0.07	919.0 ± 9.30
3	32.3 ± 1.04	5.92 ± 0.09	875.7 ± 10.10
4	41.6 ± 0.06	6.32 ± 0.04	886.0 ± 2.50
5	48.8 ± 0.86	6.29 ± 0.03	890.0 ± 4.20

Note: Available chlorine concentration (ACC). Pondus hydrogenii (pH). Oxidation reduction potential (ORP).

**Table 5 foods-14-02754-t005:** Sensory evaluation for large yellow croakers.

Indicators	Evaluation Criteria			
	10–8	8–6	6–4	4–2
Appearance	Shiny color, abdomen golden-yellow	Shiny color, abdomen golden-yellow still obvious	Slightly dim color, abdomen golden-yellow fades	Relatively dim color, abdomen golden-yellow fades significantly
Flavor	Strong fish aroma, rich umami taste, no off odor	Relatively strong fish aroma, relatively rich umami taste, no off odor	Light fish aroma, light umami taste, with fishy and off odor	No fish aroma, no umami taste, strong fishy and off odor
Morphology	Firm meat without looseness	Relatively firm meat	Slightly loose meat	Obviously loose meat
Texture	The depression caused by finger pressure recovers quickly	Depression recovers relatively quickly after finger pressing	Depression recovers slowly after finger pressing	Depression recovers slowly after finger pressing

Note: The evaluation criteria and indices presented in this table are developed with reference to the following national standards: GB/T 18108-2024, SC/T 3101-2010, and SC/T 3216-2016 [25,26,27].

**Table 6 foods-14-02754-t006:** Sensory evaluation scores of large yellow croakers by 8 assessors in different time.

Group—Storage Time (d)	Sensory Evaluation Scores								Mean ± SD
	1	2	3	4	5	6	7	8	
CG—0d	10	10	9.5	9.5	10	10	9.5	10	9.81 ± 0.26
SAEW-NB—0d	9	10	9.5	10	9.5	10	9.5	10	9.69 ± 0.37
CG—2d	8	8.5	9.5	8.5	9	9	8	9	8.69 ±0.53
SAEW-NB—2d	9	9	9.5	9	9	9	9	9	9.06 ± 0.18
CG—4d	6	7	7	7.5	7	7.5	7	6	6.88 ± 0.58
SAEW-NB—4d	9	8	8.5	9	8.5	8.5	9	9	8.63 ± 0.35
CG-6d	7	7	6.5	6	6	6	6.5	6.5	6.44 ± 0.42
SAEW-NB—6d	7.5	8	7.5	8	7.5	8	8	8	7.81 ± 0.26
CG-8d	7	7	7.5	7	7	7	6.5	7.5	5.44 ± 0.50
SAEW-NB—8d	6	6	6	5	5	5.5	5	5	7.06 ± 0.32
CG—10d	5	4	4	3	3.5	4.5	4.5	4	4.06 ± 0.62
SAEW-NB—10d	6.5	7	6	6.5	7	7	6.5	7.5	6.75 ± 0.46

Note: Control group (CG). Slightly acidic electrolyzed water combined with nano-bubbles (SAEW-NB). Days (d). Standard deviation (SD).

**Table 7 foods-14-02754-t007:** Effect of SAEW-NB treatment on the TVC and spoilage bacteria of the fish.

Indexes (log_10_ CFU/g)	Groups	Storage Time (d)					
		0	2	4	6	8	10
TVC	CG	4.13 ± 0.02	5.31 ± 0.01	6.08 ± 0.02	6.39 ± 0.09	6.69 ± 0.12	7.02 ± 0.06
	SAEW-NB	2.94 ± 0.01	3.92 ± 0.02	4.94 ± 0.06	5.38 ± 0.06	5.97 ± 0.04	6.39 ± 0.01
H_2_S-producing bacteria	CG	1.5 ± 0.28	4.26 ± 0.01	5.00 ± 0.06	5.54 ± 0.07	5.95 ± 0.07	6.27 ± 0.18
	SAEW-NB	<1	3.27 ± 0.02	4.55 ± 0.01	4.93 ± 0.05	5.25 ± 0.07	5.47 ± 0.07
*Pseudomonas*	CG	3.29 ± 0.02	4.31 ± 0.01	4.99 ± 0.06	5.70 ± 0.01	5.69 ± 0.17	5.64 ± 0.09
	SAEW-NB	2.14 ± 0.20	3.06 ± 0.03	4.18 ± 0.01	4.70 ± 0.02	5.09 ± 0.07	5.40 ± 0.14
*Enterobacteriaceae*	CG	3.75 ± 0.05	4.56 ± 0.03	4.81 ± 0.01	5.44 ± 0.01	5.94 ± 0.03	5.78 ± 0.04
	SAEW-NB	2.02 ± 0.03	2.84 ± 0.01	3.98 ± 0.02	4.45 ± 0.04	4.93 ± 0.03	5.23 ± 0.11

Note: Total viable counts (TVC). Control group (CG). Slightly acidic electrolyzed water combined with nano- bubbles (SAEW-NB). Days (d).

**Table 8 foods-14-02754-t008:** Qualitative analysis of flavor compounds in large yellow croaker meat during different storage period.

Class	Compound	CAS #	Formula	MW	RI	Rt [sec]	Dt [a.u.]
	Ethyl 2-hydroxy-4-methylpentanoate	C10348477	C_8_H_16_O_3_	160.2	1065.6	431.15	1.31
	Butyl 2-methylbutanoate	C15706737	C_9_H_18_O_2_	158.2	1232.3	715.39	1.37
	Menthyl acetate	C89485	C_12_H_22_O_2_	198.3	1289.6	819.65	1.24
	Methyl Salicylate	C119368	C_8_H_8_O_3_	152.1	1169.7	603.09	1.15
Ester(10)	Methyl isovalerate	C556241	C_6_H_12_O_2_	116.2	1016.1	372.88	1.19
	Ethyl hexanoate	C123660	C_8_H_16_O_2_	144.2	1215.4	687.15	1.79
	Ethyl 2-methylbutyrate	C7452791	C_7_H_14_O_2_	130.2	1023.4	380.91	1.24
	Ethyl-butyrate	C105544	C_6_H_12_O_2_	116.2	1048.3	409.81	1.19
	Ethyl Acetate	C141786	C_4_H_8_O_2_	88.1	887.0	286.73	1.33
	Propyl propanoate	C106365	C_6_H_12_O_2_	116.2	812.0	253.44	1.20
	1-Butanol	C71363	C_4_H_10_O	74.1	1141.5	548.65	1.20
	2-Butanol	C78922	C_4_H_10_O	74.1	1025.9	383.81	1.34
	1-Propanol	C71410	C_5_H_12_O	88.1	1258.1	760.66	1.26
	1-Pentanol	C71238	C_3_H_8_O	60.1	1026.8	384.83	1.09
Alcohol(9)	3-Pentanol	C584021	C_5_H_12_O	88.1	1134.4	535.73	1.41
	2-Methyl-1-butanol	C137326	C_5_H_12_O	88.1	1233.4	717.10	1.23
	2-Methyl-2-propanol	C75650	C_4_H_10_O	74.1	923.0	305.28	1.32
	2-Methyl-1-propanol	C78831	C_4_H_10_O	74.1	1104.4	484.64	1.17
	3-Methyl-2-butanol	C598754	C_5_H_12_O	88.1	690.3	207.44	1.23
	2-Butanone	C78933	C_4_H_8_O	72.1	909.7	297.66	1.24
	Cyclopentanone	C120923	C_5_H_8_O	84.1	792.2	245.31	1.34
	2-Heptanone	C110430	C_7_H_14_O	114.2	1190.7	646.95	1.26
Ketone(7)	1-Penten-3-one	C1629589	C5H8O	84.1	1016.1	372.83	1.07
	3-Hydroxy-2-butanone	C513860	C_4_H_8_O_2_	88.1	1289.8	820.21	1.06
	3-Methyl-2-pentanone	C565617	C_6_H_12_O	100.2	749.8	228.79	1.17
	4-Methyl-3-penten-2-one	C141797	C_6_H_10_O	98.1	1136.2	538.96	1.13
	Nonanal	C124196	C_9_H_18_O	142.20	1110.90	510.48	1.48
	Butanal	C123728	C_4_H_8_O	72.1	878.3	282.67	1.28
Aldehyde(6)	Hexanal	C66251	C_6_H_12_O	100.2	1100.0	477.44	1.26
	3-Methyl butanal	C590863	C_5_H_10_O	86.1	908.5	297.09	1.19
	Heptanal	C111717	C_7_H_14_O	114.2	1206.7	673.13	1.32
	2,4-Heptadienal	C5910850	C_7_H_10_O	110.2	999.4	355.10	1.18
	Acetic acid	C64197	C_2_H_4_O_2_	60.1	1489.1	1307.86	1.06
Acid(3)	Propanoic acid	C79094	C_3_H_6_O_2_	74.1	720.8	218.11	1.26
	2-Methylheptanoic acid	C1188029	C_8_H1_6_O_2_	144.2	1140.7	547.14	1.42
	Propyl benzene	C103651	C_9_H_12_	120.2	1207.0	673.47	1.18
	1-Heptene	C592767	C_7_H_14_	98.2	757.2	231.58	1.09
	Methylpyrazine	C109080	C_5_H_6_N_2_	94.1	1288.9	818.41	1.10
	Pyrrolidine	C123751	C_4_H_9_N	71.1	1001.4	357.13	1.29
	2-Ethyl-5-methylpyrazine	C13360640	C_7_H_10_N_2_	122.2	998.5	354.08	1.21
Alkane	Dimethyl disulfide	C624920	C_2_H_6_S_2_	94.2	1092.6	466.67	1.15
and	2-Methyl-3-methylthiofuran	C63012975	C_6_H_8_OS	128.2	1360.0	966.72	1.11
Heterocyclic(12)	2,5-Dimethylfuran	C625865	C_6_H_8_O	96.1	712.9	215.32	1.35
	2-Ethylfuran	C3208160	C_6_H_8_O	96.1	705.0	212.52	1.29
	Piperonal propylene glycol aceta	C61683996	C_11_H_12_O_4_	208.2	759.2	232.35	1.12
	5-Methyl-2-Furanmethanol	C3857258	C_6_H_8_O_2_	112.1	970.9	335.27	1.26
	1,2-Dimethoxyethane	C110714	C_4_H_10_O_2_	90.1	672.1	201.33	1.10

Note: The retention times and ion migration times are listed together with the compound name, CAS number, molecular formula, molecular weight (MW), reserved index (RI), retention time (Rt), and drift time (Dt).

## Data Availability

The original contributions presented in this study are included in the article. Further inquiries can be directed to the corresponding authors.

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
