# Peer review of "Effect of Slightly Acidic Electrolyzed Water Combined with Nano-Bubble Sterilization on Quality of Larimichthys crocea During Refrigerated Storage"

_foods, 2025, doi:10.3390/foods14152754_

Round 1

Reviewer 1 Report

Comments and Suggestions for Authors

Dear Authors,

I appreciate the comprehensive and well-structured manuscript addressing an innovative preservation method for Larimichthys crocea. The study is scientifically relevant, the methodology is sound, and the results are clearly presented and thoroughly discussed.
I suggest a few minor revisions to improve clarity and precision, particularly regarding terminology (e.g., “off-flavor volatiles” instead of “undesirable volatiles”), statistical detail, and refinement of long paragraphs in the results section. Please consider a light language polishing.

Lines 15–29: The abstract is concise and informative. Suggest replacing “undesirable volatiles” with “off-flavor volatiles” for more accurate terminology. Also, briefly mention the storage duration in the conclusion sentence to enhance contextualization.

Line 43: replace “corruption and deterioration” with “spoilage and deterioration” to avoid ambiguity.

Lines 73–76: Consider adding a clearer research hypothesis at the end of the introduction to strengthen the rationale.

Lines 78–90: The description of fish sourcing and handling is clear. Suggest adding the total elapsed time between capture and treatment.

Lines 91–111: Good description of the SAEW-NB system. It would be helpful to include the expected ORP range for each ACC level.

Lines 167–173: Specify the sensors used in the electronic tongue system (model, manufacturer).

Lines 188–194: Consider reporting average sensory scores per assessor and their standard deviations.

Line 204: Well-explained synergistic effect of SAEW-NB. Consider mentioning the equipment used for microbial plate counts.

Lines 363–378: Clarify that initial texture changes were mild and less impactful than long-term oxidative damage.

Lines 409–466: Excellent volatile compound analysis. To improve readability, consider breaking down and summarizing long paragraphs by compound class (e.g., aldehydes, alcohols, esters).

Lines 490–503: Strong and well-summarized conclusion. Suggest adding a practical note (e.g., “The SAEW-NB treatment extended shelf life by approximately X days under refrigerated conditions”).

Author Response

Dear Reviewers,

Thank you for your contributions to the improvement of this manuscript (foods-3767336), entitled "Effect of Slightly Acidic Electrolyzed Water Combined with Nano-Bubbles Sterilization on Quality of Larimichthys Crocea during Refrigerated Storage." In light of the Reviewers' comments and suggestions, we respond to your comments as follows. 

Reviewer #1:

I appreciate the comprehensive and well-structured manuscript addressing an innovative preservation method for Larimichthys crocea. The study is scientifically relevant, the methodology is sound, and the results are clearly presented and thoroughly discussed. I suggest a few minor revisions to improve clarity and precision, particularly regarding terminology (e.g., “off-flavor volatiles” instead of “undesirable volatiles”), statistical detail, and refinement of long paragraphs in the results section. Please consider a light language polishing.

  1. Lines 15–29: The abstract is concise and informative. Suggest replacing “undesirable volatiles” with “off-flavor volatiles” for more accurate terminology. Also, briefly mention the storage duration in the conclusion sentence to enhance contextualization.

Response: The “undesirable volatiles” has been revised to “off-flavor volatiles” in the manuscript, please check Line 25-26. Besides, the storage duration has been added in Line 27-28.

  1. Line 43: replace “corruption and deterioration” with “spoilage and deterioration” to avoid ambiguity.

Response: Many thanks for your valuable comments. It has been revised in the manuscript. Please check Line 43.

  1. Lines 73–76: Consider adding a clearer research hypothesis at the end of the introduction to strengthen the rationale.

Response: The sentence has been revised in the manuscript. Please check Line 73-79.

  1. Lines 78–90: The description of fish sourcing and handling is clear. Suggest adding the total elapsed time between capture and treatment.

Response: Appreciate your comments. It has been added in the article. Line 86-88.

  1. Lines 91–111: Good description of the SAEW-NB system. It would be helpful to include the expected ORP range for each ACC level.

Response: Thank you for the Reviewers’ comments. It has been provided in the article. Line 138-139

  1. Lines 167–173: Specify the sensors used in the electronic tongue system (model, manufacturer).

Response: We have added accordingly in the manuscript. Please check Line 181-186.

  1. Lines 188–194: Consider reporting average sensory scores per assessor and their standard deviations.

Response: According to the reviewer’s suggestion. It was added to the Table 6.

  1. Line 204: Well-explained synergistic effect of SAEW-NB. Consider mentioning the equipment used for microbial plate counts.

Response: The equipment of microbial plate counts is provided in the article. Please check Line 147-148.

  1. Lines 363–378: Clarify that initial texture changes were mild and less impactful than long-term oxidative damage.

Response: Thanks for the suggestions. We have provided relevant clarify in the manuscript, please check Line 388-396.

  1. Lines 409–466: Excellent volatile compound analysis. To improve readability, consider breaking down and summarizing long paragraphs by compound class (e.g., aldehydes, alcohols, esters).

Response: Thank you for kindly reminding us. We have been revised in the manuscript. Please check Line 438-488.

  1. Lines 490–503: Strong and well-summarized conclusion. Suggest adding a practical note (e.g., “The SAEW-NB treatment extended shelf life by approximately X days under refrigerated conditions”).

Response: According to the reviewer’s suggestion, we have added accordingly in Line 527-528.

Reviewer 2 Report

Comments and Suggestions for Authors

The research article with the title "Effect of Slightly Acidic Electrolyzed Water Combined with Nano-Bubbles Sterilization on Quality of Larimichthys Crocea during Refrigerated Storage" has 19 pages, 10 signed Figures and 6 Tables with citations of 53 References.  The article uses a template, has the usual structure, and citations are optional and not self-serving. Regarding the Scientific scope of MDPI "Foods" Journal, as well as the inclusion of the article in the Section Food "Packaging and Preservation" and the Special Issue entitled "Innovative Muscle Foods Preservation and Packaging Technologies". I have a few comments on the text and recommend fine-tuning the text in more detail during the proof-of-reading process.   Title and  Abstract: I have no comments about them.   Keywords: I recommend choosing keywords other than those in the manuscript title. The semicolon at the end is also one of the reasons for the above note about text control.   Methodology:  I recommend adding a reference to an article or methodology or standard for processing marine fish, ideally of this species, to the experimental methodology (lines 83 to 85). It would be good to specify Larimichthys Crocea using the results of basic chemical analysis, and to state the average weight or age of the fish included in the experiment. Chapter 2.2 requires the addition of a source (References - article, standard, methodology).   Line 146: What pH meter? (manufacturer, origin, type)?   Line 195: Table No. 4 would benefit from better formatting and clarity. It is also not clear where this type of assessment comes from. From the authors, or as stated in the text (to GB/T 18108-2019, SC/T 3101- 189 2010 and SC/T 3216-2016) from some standards? It should also be in the note or the title of the table.   Were these assessors trained? Do they have food assessment tests (certificate?) that their senses are in order? Were they trained to assess, what were the conditions for assessment? (according to ISO standards or not?) How was the fish prepared (temperature) before and after the samples were assessed?
Was a neutralizer used (of the senses - bread, water, alcohol, etc.)?   Results and Discussion: Figures 5 to 9 are difficult to read in the format that was chosen. Figures marked as No. 8 appear twice in the manuscript (lines 408 and 468).

Conclusion:

I have no comments.

I recommend revising and editing the article.         

Author Response

Dear Reviewers,

We are deeply appreciative of your efforts in helping to refine our manuscript (foods-3767336), whose title is "Effect of Slightly Acidic Electrolyzed Water Combined with Nano-Bubbles Sterilization on Quality of Larimichthys Crocea during Refrigerated Storage." Having considered the Reviewers' comments and suggestions, we respond to your comments as follows. 

Reviewer 2:

Keywords:

  1. I recommend choosing keywords other than those in the manuscript title. The semicolon at the end is also one of the reasons for the above note about text control.

Response: Thank you for pointing this out. Keywords are revised in the article. Please check Line 30-31.

Methodology:

  1. I recommend adding a reference to an article or methodology or standard for processing marine fish, ideally of this species, to the experimental methodology (lines 83 to 85).

Response: We appreciate your suggestion. The relevant references have been added to the manuscript. Please check Reference 17.

  1. It would be good to specify Larimichthys Crocea using the results of basic chemical analysis, and to state the average weight or age of the fish included in the experiment.

Response: Basic information and composition have been provided in Table 2. Please check Table 1.

  1. Chapter 2.2 requires the addition of a source (References-article, standard, methodology).

Response: We have added accordingly. Please check Reference 19-20.

  1. Line 146: What pH meter? (manufacturer, origin, type)?

Response: We have added accordingly. Please check Line 156.

  1. Line 195: Table No. 4 would benefit from better formatting and clarity. It is also not clear where this type of assessment comes from. From the authors, or as stated in the text (to GB/T 18108-2019, SC/T 3101- 189 2010 and SC/T 3216-2016) from some standards? It should also be in the note or the title of the table.

Response: We have provided the corresponding explanations. Please check Line 213-215.

  1. Were these assessors trained? Do they have food assessment tests (certificate?) that their senses are in order? Were they trained to assess, what were the conditions for assessment? (according to ISO standards or not?)

Response: Yes, in this study, all sensory assessors consisted of postgraduate students and faculty members with long-term research experience in Larimichthys crocea. Prior to the experiment, they received specialized training tailored to aquatic product sensory assessment, following the general principles of ISO 8586-1:2012 (Sensory analysis—Guidelines for the selection, training and monitoring of assessors). To verify their sensory competence, all assessors passed internal sensory function tests, including discrimination of basic tastes (sour, bitter, etc.) and identification of typical off-odors in aquatic products, which are aligned with the core requirements of ISO 11036:2021 (Sensory analysis—Guidelines for the assessment of sensory proficiency).

  1. How was the fish prepared (temperature) before and after the samples were assessed? Was a neutralizer used (of the senses - bread, water, alcohol, etc.)?

Response: Thanks for your professional suggestions. It has been provided in Line 205-210. Yes, water is used neutralizer in the study. Line 210.

Results and Discussion:

  1. Figures 5 to 9 are difficult to read in the format that was chosen. Figures marked as No. 8 appear twice in the manuscript (lines 408 and 468).

Response: We have modified Figures 5 to 10 format to enhance readability. And the corresponding figure numbers have been revised in the article.
